# Impact of Pain Education on Pain Relief in Oncological Patients: A Narrative Review of Systematic Reviews and Meta-Analyses

**DOI:** 10.3390/cancers17101683

**Published:** 2025-05-16

**Authors:** Erika Galietta, Costanza M. Donati, Alberto Bazzocchi, Rebecca Sassi, Arina A. Zamfir, Renée Hovenier, Clemens Bos, Nikki Hendriks, Martijn F. Boomsma, Mira Huhtala, Roberto Blanco Sequeiros, Holger Grüll, Simone Ferdinandus, Helena M. Verkooijen, Alessio G. Morganti

**Affiliations:** 1Department of Medical and Surgical Sciences, Alma Mater Studiorum University of Bologna, 40138 Bologna, Italy; erika.galietta2@unibo.it (E.G.); costanzamaria.donati@unibo.it (C.M.D.); alessio.morganti2@unibo.it (A.G.M.); 2Radiation Oncology, IRCCS Azienda Ospedaliero-Universitaria di Bologna, 40138 Bologna, Italy; 3Diagnostic and Interventional Radiology, IRCCS Istituto Ortopedico Rizzoli, 40136 Bologna, Italy; alberto.bazzocchi@ior.it (A.B.); rebecca.sassi@ior.it (R.S.); 4Division of Imaging and Oncology, University Medical Center Utrecht, Heidelberglaan 100, 3584 CX Utrecht, The Netherlands; r.hovenier-2@umcutrecht.nl (R.H.); c.bos@umcutrecht.nl (C.B.); h.m.verkooijen@umcutrecht.nl (H.M.V.); 5Department of Radiology, Isala Hospital, Dokter van Heesweg 2, 8025 AB Zwolle, The Netherlands; n.hendriks@isala.nl (N.H.); m.f.boomsma@isala.nl (M.F.B.); 6Department of Oncology and Palliative Center, Turku University Hospital, University of Turku, 20520 Turku, Finland; mira.m.huhtala@varha.fi; 7Research Unit of Medical Imaging, Physics and Technology, Faculty of Medicine, University of Oulu, 90014 Oulu, Finland; roberto.blanco@varha.fi; 8Department of Radiology, Turku University Hospital, Kiinamyllynkatu 4-8, 20520 Turku, Finland; 9Institute of Diagnostic and Interventional Radiology, Faculty of Medicine, University Hospital Cologne, University of Cologne, 50923 Cologne, Germany; holger.gruell@uk-koeln.de; 10Department of Radiation Oncology, Cyberknife and Radiotherapy, Faculty of Medicine and University Hospital Cologne, 50931 Cologne, Germany; simone.ferdinandus@uk-koeln.de; 11Center for Integrated Oncology (CIO), Faculty of Medicine and University Hospital, University of Cologne, 50923 Cologne, Germany; 12Division Imaging, Antoni van Leeuwenhoek, Plesmanlaan 121, 1066 CX Amsterdam, The Netherlands

**Keywords:** pain education, cancer pain, narrative review, systematic review, meta-analysis, pain management, patient education, quality of life, medication adherence

## Abstract

Pain is a common and challenging symptom for cancer patients, and despite advances in treatment, it often remains inadequately managed. One approach to improving pain control is pain education (PE), which aims to increase patients’ understanding of pain management. However, studies on the effectiveness of PE programs have shown mixed results. This review analyzed existing research on PE in cancer patients, examining systematic reviews and meta-analyses. While some studies found PE to significantly reduce pain, others showed no improvement. However, PE consistently helped patients understand pain better and adhere to medications. The variation in outcomes could be due to differences in pain types, clinical settings, and study designs. Further research is needed to clarify how PE can be most effectively applied to improve pain management in cancer patients.

## 1. Introduction

Pain is a complex and burdensome symptom experienced by oncological patients and their caregivers, despite advances in guidelines and novel approaches in pain management. The prevalence of pain among cancer patients remains alarmingly high, with recent systematic literature reviews and meta-analyses reporting an overall prevalence of 44.5% and a 30.6% prevalence of moderate to severe pain intensity [1].

This persistent challenge in pain control can be attributed to various barriers, with knowledge gaps among healthcare professionals and patients being among the primary factors [2]. Healthcare providers face challenges in identifying, assessing, and managing pain, while fear of opioid prescriptions, including concerns about tolerance, addiction, and adverse effects, hinders effective pain management. Similarly, patients’ beliefs about pain being uncontrollable and fears surrounding opioid use are significant barriers to optimal pain relief [3,4].

To address these barriers and improve pain management, pain education (PE) programs have emerged as a promising intervention for both cancer patients and healthcare professionals. Pain education in oncology is delivered through face-to-face sessions, telephone interventions, and integrated multimodal programs that may include printed materials, audiovisual content, or digital platforms [5,6,7,8,9,10,11,12,13,14,15,16]. These interventions aim to improve patients’ understanding of pain mechanisms, treatment options, and self-management strategies. Although no universal standard exists, common components include information about analgesics (especially opioids), addressing fears and misconceptions, and fostering adherence and self-efficacy. Some programs also incorporate interactive training in pain control techniques. However, despite the wealth of research, the conclusions drawn from these reviews have been varied and heterogeneous. Two fundamental questions remain unanswered: firstly, given the increasing knowledge and awareness about pain, is PE truly effective in alleviating patients’ symptoms? Secondly, does the effectiveness of PE depend on the delivery methods used? Are certain methods more effective than others?

In light of these uncertainties, we conducted a comprehensive review of the related literature to address these open questions. We focused on literature reviews based on the AMSTAR-2 (A MeaSurement Tool to Assess systematic Reviews) guidelines [17], ensuring a standardized approach to the review process. By analyzing the existing evidence on PE, our aim is to provide insights into its true effectiveness and explore the impact of different educational delivery methods.

## 2. Materials and Methods

### 2.1. Literature Search

This narrative review of the literature was conducted by a multidisciplinary team comprising palliative care specialists, radiation oncologists, radiologists, and experts in image-guided treatment for bone metastases. Only systematic reviews and/or meta-analyses published in English and specifically addressing pain education interventions in patients with cancer were considered eligible. Reviews on non-cancer pain or those not evaluating pain education as a primary intervention were excluded. PubMed, Scopus, and the Cochrane Library were used as the bibliographic databases. The SANRA (scale for the quality assessment of narrative review articles) guidelines [18] were followed to ensure a systematic and comprehensive approach. The original search was conducted on 20 August 2023 and updated on 9 January 2025. No time limits were applied in terms of publication year. The search strategy employed in PubMed involved using the keywords “pain education” AND (“cancer” OR “tumor” OR “oncolog*”) AND “review” with the filter “Review” applied. To ensure comprehensive coverage and minimize the risk of omitting relevant studies, we intentionally adopted a broad search strategy without exclusively relying on specific terms such as “systematic review”, “meta-analysis”, or “meta-analytic”. However, during the selection phase, only systematic reviews and meta-analyses were included based on the predefined eligibility criteria. Additionally, the snowball technique was used to extend the search by exploring the reference lists of relevant articles.

### 2.2. Narrative Review Checklist

A narrative review checklist was followed to ensure comprehensive coverage of the topic. Appendix A provides details of the checklist items and their corresponding assessment criteria.

### 2.3. Data Analysis

Given the substantial heterogeneity observed among the studies included in this review in terms of selection criteria, endpoints, and analyzed parameters, a quantitative meta-analysis was deemed inappropriate. Due to this heterogeneity, we did not conduct a statistical synthesis or apply formal meta-analytic techniques. Instead, we opted for a narrative approach to summarize and critically interpret the available evidence. We did not extract or re-evaluate individual primary studies included in each systematic review or meta-analysis. Rather, we analyzed and synthesized each review in its entirety, as per narrative review methodology. This approach allowed us to provide a structured and qualitative overview of the evolution of pain education in oncology care, based on higher-level synthesized evidence.

### 2.4. Quality Assessment

The quality assessment of the studies included in this review was independently performed by two authors (MB and CMD) using the AMSTAR-2 tool [17]. The AMSTAR-2 tool is a validated instrument for evaluating the methodological quality of systematic reviews. Although formal methods to assess publication bias (e.g., funnel plots) were not applicable due to the narrative design of this review, the included meta-analyses differed in whether they reported such assessments. The potential for publication bias in the primary studies remains a limitation and is reflected in the AMSTAR-2 evaluations. We refer readers to Appendix A for further detail. Each study was assessed for critical and non-critical flaws, with the overall confidence rating based on the AMSTAR-2 guidelines categorized as follows: “high” confidence: assigned to studies with 0–1 non-critical weakness; “moderate” confidence: assigned to studies with more than 1 non-critical weakness; “low” confidence: assigned to studies with 1 critical flaw with/without non-critical weaknesses; “critically low” confidence: assigned to studies with more than 1 critical flaw with/without non-critical weaknesses.

## 3. Results

### 3.1. Search Results

A total of 54 records were initially identified based on the selection criteria. After eliminating duplicate records and papers not considered relevant for the specific objectives of this review, 12 publications were selected. Furthermore, three papers were excluded from this analysis as they reported narrative reviews of the literature [5,6,7]. Finally, six systematic reviews [8,9,10,11,12,13] and three meta-analyses [14,15,16] were included in this review (Figure 1).

The methodologies varied across reviews: systematic reviews included randomized controlled trials (RCTs), quasi-experimental studies, pre-post designs, and retrospective reviews, evaluating educational interventions for patients and healthcare providers across multiple clinical settings such as hospitals, outpatient units, home, and hospice care [8,9,10,11,12,13]. The number of studies included ranged from 2 to 26, highlighting variability in the scope and inclusion criteria. The three meta-analyses exclusively analyzed RCTs, assessing the efficacy of educational interventions on pain intensity, pain management knowledge, and patient adherence. These meta-analyses involved between 11 and 12 studies each, collectively examining the effects on thousands of cancer patients. Despite differences in intervention settings and educational strategies, all reviews consistently reported variability in the outcomes, particularly in the effectiveness of these interventions in reducing pain severity and improving quality of life [14,15,16].

### 3.2. Quality Assessment

The quality assessment of the selected publications is presented in Appendix A. One review was classified as having a high overall confidence rate [15], one review was categorized as having a low overall confidence rate [13], and seven reviews were classified as having a critically low overall confidence rate [8,9,10,11,12,14,16]. Upon further examination, the AMSTAR-2 domain with the highest number of critical weaknesses across the selected reviews was the “availability of a priori protocol.”

### 3.3. Effectiveness of Pain Education on Pain Relief

#### 3.3.1. Systematic Reviews

Three systematic reviews consistently reported a significant reduction in pain intensity after PE in at least half of the studies analyzed [8,9,10]. Conversely, another review indicated a significant reduction only in usual and average pain intensity [11]. However, two reviews showed a significant reduction in pain intensity in less than half of the studies analyzed [12,13], and one review included a randomized clinical trial showing increased, worse pain after PE [11]. Additionally, one review evaluated the impact of PE on the interference of pain in daily activities, reporting an improvement in this parameter in four out of the 12 studies analyzed [13] (Table 1 and Table 2).

#### 3.3.2. Meta-Analyses

Two meta-analyses reported a statistically significant but overall small reduction in pain intensity after PE [14,16]. Conversely, one meta-analysis found no improvements in overall and current pain intensity but recorded a reduction only in least pain intensity [15]. Moreover, two meta-analyses showed no beneficial effect [16] or reduced effect of PE in randomized trials using attention control as a control group [14] (Table 3 and Table 4).

#### 3.3.3. Other Effects of Pain Education

Several reviews evaluated the impact of PE on pain knowledge, and in all cases, a significant improvement was observed in most of the studies analyzed [8,10,13]. Furthermore, two systematic reviews assessed the impact of PE on medication adherence, with a significant improvement reported in half of the studies analyzed [10,13], while a meta-analysis recorded this improvement in only one out of three studies [14]. Finally, a systematic review analyzing four randomized trials evaluating the impact of PE on quality of life (QoL) did not report an improvement in this parameter in any of the reports evaluated [11].

#### 3.3.4. Effectiveness of Different Pain Education Models

Several studies examined differences in effectiveness based on the format or intensity of educational interventions. For example, some reviews and meta-analyses found that single-session and multiple-session formats produced similar effects on pain intensity [8,14], while other studies observed greater benefits from high-dose or high-intensity education programs (defined as ≥2 h per session or ≥4 total sessions, often involving dedicated personnel) [15]. In contrast, no single intervention component (e.g., structured vs. tailored) consistently predicted better outcomes, and even the longest interventions did not guarantee improved results [12]. Interestingly, the poorest outcomes were reported in a study using a very short telephone-only session. These findings are summarized in Table 5 and suggest that both content and delivery method play a role in the effectiveness of pain education.

## 4. Discussion

### 4.1. Narrative

The present narrative review integrates evidence from six systematic reviews and three meta-analyses evaluating pain-education (PE) programs for adult oncology populations. Two conclusions are well supported across the corpus. First, PE consistently enhances disease-specific knowledge and improves adherence to prescribed analgesic regimens [8,9,10,13,14]. Second, these cognitive and behavioral gains do not translate into uniform, clinically important reductions in pain intensity. The meta-analysis with the highest methodological quality (AMSTAR-2 “high”) detected no significant change in overall or current pain intensity and only a modest improvement in least pain intensity (mean difference −0.25 on an 11-point scale) [15]. Earlier meta-analyses did report statistically significant reductions, yet the absolute effect sizes were small (standardized mean difference −0.11 to −0.18) and disappeared when pain education was compared with attention-control interventions rather than usual care [14,16]. These observations imply that the therapeutic benefit frequently ascribed to educational content may in fact reflect the non-specific advantages of structured patient engagement.

### 4.2. Heterogeneity of Intervention Characteristics and Outcomes

Interventions varied along multiple dimensions, including duration (15 min to >4 h per session), number of contacts, delivery modality (face-to-face, telephone, multimedia), and personnel involved. Cummings et al. documented a dose–response association, whereby programs delivering ≥2 h per session or ≥4 sessions were more likely to demonstrate analgesic benefit, whereas the briefest, telephone-only intervention proved least effective [15]. Conversely, Koller et al. were unable to relate efficacy to any single component, suggesting that intervention intensity alone is insufficient to explain variability in outcomes [12]. Outcome heterogeneity further complicates interpretation; at least six discrete pain constructs—overall, average, worst, least, usual, and current pain—were employed across reviews [11,14,15,16]. Such fragmentation impedes meta-analysis and limits cross-study comparability.

### 4.3. Influence of Comparator Selection

Trials employing usual-care comparators tended to report favorable effects, whereas those using structurally equivalent attention controls did not [14,16]. This pattern indicates that empathic attention, reinforcement of self-management, and other non-specific elements of the clinical encounter may constitute active ingredients in PE interventions. Disentangling specific educational content from contextual effects remains a critical research priority.

### 4.4. Methodological Quality of the Evidence Base

Seven of nine reviews received a “critically low” AMSTAR-2 rating owing to absent protocols, single-reviewer selection, or inadequate assessment of publication bias [8,9,10,11,12,14,16]. Three reviews incorporated non-randomized studies [8,12,14], thereby introducing potential confounding typical of observational research and risking the phenomenon of “spurious precision” described by Egger et al. [19]. Even among randomized controlled trials, allocation concealment was frequently unclear, outcome blinding was rare, and follow-up seldom exceeded eight weeks; consequently, the durability of any observed benefit cannot be asserted with confidence.

### 4.5. Clinical Implications

The robust improvements in pain knowledge and medication adherence warrant the incorporation of structured PE into routine oncology and palliative-care pathways, particularly where misconceptions regarding opioid therapy remain pervasive. Nevertheless, when rapid, clinically meaningful analgesia is required—for example, in patients experiencing moderate-to-severe cancer pain—PE should be regarded as adjunctive rather than substitutive. Recent primary evidence supports this view: a nurse-led educational intervention combined with telephone follow-up accelerated and amplified the analgesic effect of palliative radiotherapy for bone metastases, presumably by fostering adherence and timely titration of rescue medication [20]. Whether comparable benefit will accompany emerging loco-regional modalities such as MR-guided high-intensity focused ultrasound is under prospective evaluation [21].

### 4.6. Limitations and Gaps in Current Knowledge

Short follow-up periods preclude assessment of long-term analgesic sustainability, functional recovery, or quality-of-life (QoL) trajectories. Self-reported adherence introduces social-desirability bias, and economic endpoints are seldom captured, limiting cost-effectiveness inference. Equity dimensions remain largely unexplored; preliminary evidence suggests that multimodal education may yield greater pain-related gains in women [22], whereas sociocultural norms encouraging stoicism may lead men to under-report pain, thereby attenuating measurable intervention effects [23]. No review stratified outcomes by sex, ethnicity, or socioeconomic status.

### 4.7. Future Research Directions

Methodologically rigorous trials are required to identify the minimum effective “dose” of PE and to delineate the incremental benefit of educational content relative to non-specific clinician attention. Factorial designs separating these components would be particularly informative. Longer follow-up is necessary to establish persistence of analgesic benefit and impact on broader patient-centered outcomes, including QoL, functional capacity, and healthcare utilizations. Embedded economic evaluations will aid policy makers in resource allocation, while prospective subgroup analyses should address gender, cultural, and socioeconomic modifiers of treatment response. In addition, conversational agents and other artificial-intelligence chatbots represent an emerging modality for personalized pain education; rigorous randomized trials and subsequent systematic reviews are needed to determine their comparative effectiveness and cost utility in oncological settings.

## 5. Summary

Pain-education programs in oncology reliably improve patient knowledge and medication adherence; however, current evidence supports only a modest and context-dependent reduction in pain intensity, with negligible impact on QoL. Apparent analgesic benefits diminish when educational content is isolated from structured patient engagement, underscoring the role of therapeutic alliance. Until large, well-controlled trials define optimal format, dosage, and target populations, PE should be implemented as an adjunct within multidisciplinary pain-management strategies [24] rather than as a stand-alone analgesic intervention.

## 6. Conclusions

In the aggregate evidence, every pain-education modality examined (individual nurse counselling, multidisciplinary classroom sessions, structured printed booklets, telephone reinforcement, web- or video-based modules, and multimodal combinations) produced a measurable gain in pain-related knowledge or beliefs in at least two-thirds of the primary studies reviewed [8,9,10,13,14,15,16]. No modality, however, emerged as consistently more effective than its peers once differences in intervention dose, contact time, and choice of comparator were taken into account. High-dose or multisession programs occasionally yielded larger absolute gains, but head-to-head trials are sparse and heterogeneous, precluding a definitive hierarchy of tools. Current data therefore support the conclusion that the choice of educational vehicle should be governed by feasibility, patient preference, and resource availability rather than by an assumed intrinsic superiority of any single format. Well-designed, adequately powered comparative studies remain necessary to establish whether one delivery method can meaningfully outperform another in improving both pain awareness and downstream clinical outcomes.

## Figures and Tables

**Figure 1 cancers-17-01683-f001:**
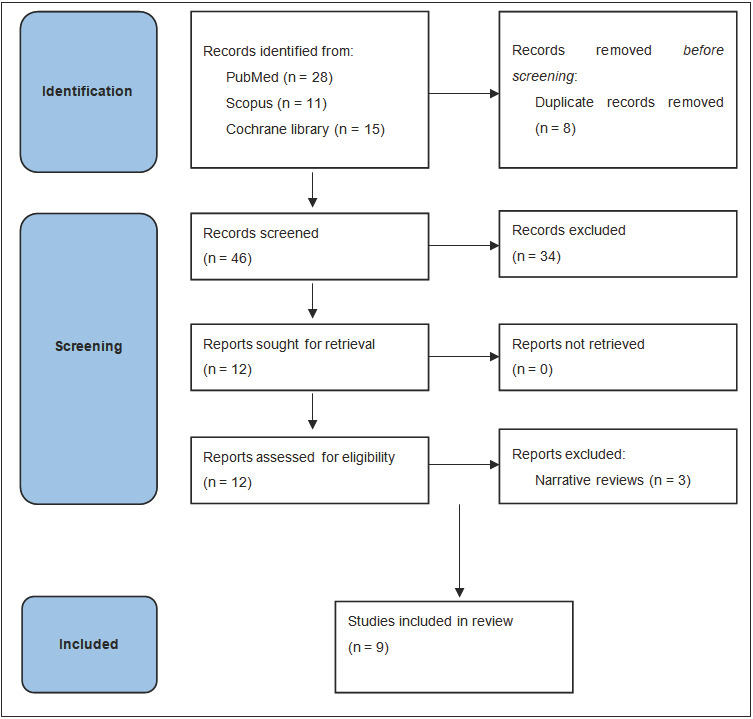
Flow diagram of study selection process.

**Table 1 cancers-17-01683-t001:** Characteristic of systematic reviews.

Reference (Author, Year)	Objective of Review	Participants/Population	No. of Studies Included	Study Design Focus	Pain Education Intervention	Pain Intensity Concept/Tool
Allard et al., 2001 [8]	Evaluate the effect of a nurse-delivered educational intervention on cancer pain management.	Cancer patients receiving nurse-based PE	Systematic review (2 RCTs and 6 nRCTs)	RCTs	Single vs. multiple home nurse visits	NRS (self-reported pain)
Goldberg et al., 2007 [9]	Assess effectiveness of educational interventions in improving cancer pain outcomes.	Cancer patients (various settings)	Systematic review (2 RCTs)	RCTs and non-RCTs	Various formats: printed, face-to-face, multimedia	Varied, often NRS or VAS
Oldenmenger et al., 2009 [10]	Identify barriers hindering adequate cancer pain management and critically review interventions to overcome these barriers.	Cancer patients (various clinical settings, mixed inpatient/outpatient)	Systematic review (10 RCTs)	Mixed methods (RCT and non-RCTs)	Patient education programs (single to multiple sessions)	Various: average, current, worst pain (often NRS 0-10)
Ling et al., 2012 [11]	Evaluate effects of psychoeducational interventions in cancer pain.	Cancer patients (all stages)	Systematic review (4 RCTs)	RCTs	Psychoeducational models	Average pain via NRS
Koller et al., 2012 [12]	Assess efficacy of various pain management education strategies.	Cancer patients (mixed stages, global)	Systematic review (23 RCTS and 1 nRCT)	RCTs	Structured vs. tailored interventions	NRS/VAS (not consistently defined)
Oldenmenger et al., 2018 [13]	Update and expand previous review on pain education in cancer.	Cancer patients receiving PE	Systematic review (26 RCTs)	Mixed	Multimodal pain education	Multiple scales

Legend: CI: confidence interval; NRS: Numeric Rating Scale; VAS: Visual Analog Scale; PE: pain education; RCT(s): randomized controlled trial(s); nRCT(s): non-randomized controlled trial(s); SMD: standardized mean difference.

**Table 2 cancers-17-01683-t002:** Results of systematic reviews.

Authors/Year [ref]	Effect of Pain Education on Pain	Other Findings
Allard et al./2001 [8]	Improved pain relief (3/4 studies).	Improved knowledge about or attitude towards cancer pain (6/6 studies).
Goldberg et al./2007 [9]	Improved pain scores (1 study: decreased PI recorded only in patients not requiring home nursing).	Reduced negative pain beliefs and misconception.
Oldenmenger et al./2009 [10]	Improvement in pain relief in 5/10 studies (“clinically significant” † only in 2 studies). Less pain in the control group in 3 studies (p: NS).	Improved: knowledge about cancer pain and its management (8/10 studies); long-term understanding of cancer pain (1/3 studies); medication adherence (3/6 studies)
Ling et al./2011 [11]	Reduced usual PI (1/4 RCTs) and average PI (1/4 RCTs); unchanged average PI (1/4 RCTs); increased worst PI (1/4 RCTs).	No change in terms of QoL (4/4 RCTs)
Koller et al./2012 [12]	Clinically meaningful effect in 9/19 evaluable studies	NR
Oldenmenger et al./2018 [13]	Statistically significant reduction in: PI (8/26 RCTs; 0/7 published after 2011) pain interference (4/12 RCTs)	improved: knowledge about cancer-related pain/pain barriers (15/22 RCTs) medication adherence (3/6 RCTs)

Legend: NR: not reported; nRCTs: non-randomized clinical trials; PI: pain intensity; QoL: quality of life; RCTs, randomized clinical trials; †: 30% reduction in pain intensity or ≥2 point reduction on a 11 points scale.

**Table 3 cancers-17-01683-t003:** Characteristic of meta-analyses.

Reference (Author, Year)	Objective of Review	Participants/Population	No. of Studies Included	Study Design Focus	Pain Education Intervention	Pain Intensity Concept/Tool
Bennett et al., 2009 [14]	Quantitatively analyze impact of PE on cancer pain using meta-analysis.	RCTs with cancer patients	Meta-analysis (10 RCTs and 2 nRCTs)	RCTs	PE session vs. control	SMDs from RCTs
Cummings et al., 2011 [15]	Review evidence of PE on cancer-related pain intensity.	Cancer patients in RCTs	Meta-analysis (11 RCTs)	RCTs	PE intensity/duration comparisons	SMD, mean difference
Jho et al., 2013 [16]	Summarize effectiveness of PE interventions on pain and related outcomes.	Adult cancer patients	Meta-analysis (12 RCTs)	RCTs	General PE strategies	Not standardized across studies

Legend: CI: confidence interval; NRS: Numeric Rating Scale; VAS: Visual Analog Scale; PE: pain education; RCT(s): randomized controlled trial(s); SMD: standardized mean difference.

**Table 4 cancers-17-01683-t004:** Results of meta-analyses.

Authors/ Year [Ref]	Effect of Pain Education on Pain	Other Findings
Bennett et al./2009 [14]	Small but statistically significant improvement in average, worst or maximum, least, and current PI. SMD −0.11 (95% CI −0.20 to −0.02) No significant differences in terms of pain interference with daily life. Higher improvement in PI recorded in studies with usual care and not with placebo/attention as comparators. Improvement of the average pain recorded in both types of comparison.	Medication adherence improved in 1/3 studies.
Cummings et al./2011 [15]	NS impact on pain interference, overall pain, and current/present pain. SMD −0.12 (95% CI −0.21 to −0.04) Significant improvement in least PI (evaluated in 2 RCTs).	NR
Jho et al./2013 [16]	Overall, small reduction in pain intensity (standardized mean difference: −0.11 (95% CI: −0.2 to −0.02) No beneficial effects in; high-quality RCTs; RCTs using attention control as a control group; Trials with 1st follow-up at ≥2 weeks.	NR

Legend: NR: not reported; nRCTs: non-randomized clinical trials; NS: not statistically significant; PI: pain intensity; RCTs, randomized clinical trials; SMD: standardized mean difference.

**Table 5 cancers-17-01683-t005:** Comparisons among different pain education strategies.

Authors/ Year [Ref]	Comparative Results
Allard et al./2001 [8]	A single brief nurse EI is effective as a series of 3 EI based on home visits
Bennett et al./2009 [14]	Single session EIs: equally effective as multiple EIs in reducing the maximum PI Differences in other outcomes not evaluable.
Cummings et al./2011 [15]	Significant effect on PI: High EI dose *: 7/9 RCTs; low EI dose †: 6/17 RCTs; p: 0.039 High EI intensity ‡: 8/10 RCTs; low EI intensity §: 5/15 RCTs; p: 0.022.
Koller et al./2012 [12]	Effect on PI: Not correlated with EI single components or their patterns, number of sessions, and type (structured vs tailored) Not improved in the 2 studies with longest EI (180–272 min) Reduced in the study with the shortest EI, provided via telephone.

Legend: EI: education intervention; PI: pain intensity; *: ≥2 of the following: follow-up ≥ 1 months, ≥2 h of education in a single session or ≥4 education sessions, significant resource allocation (dedicated staff, multidisciplinary team, other); †: intervention dose that cannot be defined as “high intervention dose”, ‡: ≥2 h of education in a single session or ≥4 education sessions; §: Intervention intensity that cannot be defined as “high intervention intensity”.

## Data Availability

Data supporting reported results can be found at Radiotherapy Unit—A.G. Morganti of the IRCCS Azienda Ospedaliero-Universitaria di Bologna.

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
