# Peer review of "Impact of Pain Education on Pain Relief in Oncological Patients: A Narrative Review of Systematic Reviews and Meta-Analyses"

_cancers, 2025, doi:10.3390/cancers17101683_

Round 1

Reviewer 1 Report

Comments and Suggestions for Authors The paper seems well written and complete and in my opinion it can be considered for publication.
I only have a few requests for further information: 1. The authors should better describe the methods of selecting the studies by better specifying
the inclusion criteria 2. Which meta-analytic methods were used in the analyses? 3. Can the results be summarized with the classic graphic representations
used in meta-analysis?

Author Response

REVIEWER 1

Comment 1

The paper seems well written and complete and in my opinion it can be considered for publication. I only have a few requests for further information. The authors should better describe the methods of selecting the studies by better specifying the inclusion criteria.

Response 1:

We sincerely thank the reviewer for the positive feedback and constructive suggestion. We recognize that our inclusion criteria could be described more clearly. To address this, we have slightly revised the paragraph in the Literature Search subsection to explicitly state that only systematic reviews and meta-analyses focused on pain education in cancer patients were considered eligible.

Revised text (replacing a sentence):

“Only systematic reviews and/or meta-analyses published in English and specifically addressing pain education interventions in patients with cancer were considered eligible. Reviews on non-cancer pain or those not evaluating pain education as a primary intervention were excluded.”

Position:
Section 2: Materials and Methods – Literature Search paragraph, replacing the sentence: “Only papers reporting systematic reviews and/or meta-analyses and published in English were included in this review.”

Comment 2

Which meta-analytic methods were used in the analyses?

Response 2:

We thank the reviewer for this important question and apologize for not having made the nature of our review more explicit. As outlined in the Data Analysis subsection, our review was conducted using a narrative approach due to the marked heterogeneity among the included studies in terms of design, interventions, and outcome measures. To make this clearer, we have expanded the explanation in the Data Analysis paragraph.

Revised text (adding a clarification):

“Due to this heterogeneity, we did not conduct a statistical synthesis or apply formal meta-analytic techniques. Instead, we opted for a narrative approach to summarize and critically interpret the available evidence.”

Position:
Section 2: Materials and Methods – Data Analysis paragraph, added after: “Given the substantial heterogeneity observed among the studies included in this review in terms of selection criteria, endpoints, and analyzed parameters, a quantitative meta-analysis was deemed inappropriate.”

Comment 3

Can the results be summarized with the classic graphic representations used in meta-analysis?

Response 3:

We are grateful to the reviewer for this suggestion. However, as explained above, our review was intentionally designed as a narrative synthesis due to the considerable heterogeneity across studies. Therefore, traditional meta-analytic visualizations such as forest plots were not applicable. We kindly refer the reviewer to our response to Comment 2 for a more detailed explanation.

No manuscript changes were required for this point, as it is addressed by the clarification added under Comment 2.

Reviewer 2 Report

Comments and Suggestions for Authors

I was very pleased to read the review by respected Erika Galietta and co-authors. The review is devoted to the impact of pain education on pain relief in oncological patients. The topic raised by the authors is of great relevance, since severe pain syndrome accompanies many oncological diseases. Pain treatment means not only improving the quality of life, but also improving the prognosis of the disease, since pain is realized in complex pathogenetic pathways that affect the adaptive potential of the body. The strengths of the review are the richness of the material and its practical value. The review is clearly structured, consists of a discussion of systematic reviews and meta-analyses. The review contains three summary tables, as well as supplementary material. The review fully complies with the stated goal, and is quite complete. Over the past ten years and earlier, no similar narrative review has been published, summarizing the results of a number of systematic reviews and meta-analyses. Much more common are works on musculoskeletal pain in neuroscience. As for pain education in cancer patients, there are not many studies.

The review includes 21 sources, starting from 1997 and ending in 2023. Thus, the review covers a fairly wide time interval of publications. The conclusions made by the authors are correct and supported by references. The prospects of the work are also described in great detail, and this study definitely requires continuation.

I highly appreciate the article of the authors, but I missed a few points that I hope the authors will take into account.

  1. I missed information about what pain education actually includes. The authors mention in the introduction that patients are told about the possibility of controlling pain and about the drugs used. I would like the authors to describe this point more clearly. Are there any standardized pain education protocols? Does it include pain control training or only educational information? 
  2. Is there any information on gender differences in relation to pain education? Do men or women show better results than the general population?
  3. And a minor comment. It seems to me that in the conclusion it is not worth citing references and discussing them. Perhaps lines 308 to 320 should be moved to the discussion section.

Author Response

REVIEWER 2

Comment 1:

I was very pleased to read the review by respected Erika Galietta and co-authors. The review is devoted to the impact of pain education on pain relief in oncological patients. The topic raised by the authors is of great relevance, since severe pain syndrome accompanies many oncological diseases. Pain treatment means not only improving the quality of life, but also improving the prognosis of the disease, since pain is realized in complex pathogenetic pathways that affect the adaptive potential of the body. The strengths of the review are the richness of the material and its practical value. The review is clearly structured, consists of a discussion of systematic reviews and meta-analyses. The review contains three summary tables, as well as supplementary material. The review fully complies with the stated goal, and is quite complete. Over the past ten years and earlier, no similar narrative review has been published, summarizing the results of a number of systematic reviews and meta-analyses. Much more common are works on musculoskeletal pain in neuroscience. As for pain education in cancer patients, there are not many studies.

Response 1:

We are deeply grateful to the reviewer for this generous and thoughtful comment. We sincerely appreciate your recognition of the relevance, completeness, and structure of our review, as well as your acknowledgment of its novelty and practical value. Your positive feedback is greatly appreciated and encouraging.

No changes to the manuscript were required in response to this comment.

Comment 2:

The review includes 21 sources, starting from 1997 and ending in 2023. Thus, the review covers a fairly wide time interval of publications. The conclusions made by the authors are correct and supported by references. The prospects of the work are also described in great detail, and this study definitely requires continuation.

Response 2:

We thank the reviewer for this positive and supportive evaluation. We are glad that the time span of the literature included, as well as the conclusions and future perspectives, were found appropriate and well-grounded.

No changes to the manuscript were required in response to this comment.

Comment 3:

I highly appreciate the article of the authors, but I missed a few points that I hope the authors will take into account. I missed information about what pain education actually includes. The authors mention in the introduction that patients are told about the possibility of controlling pain and about the drugs used. I would like the authors to describe this point more clearly. Are there any standardized pain education protocols? Does it include pain control training or only educational information?

Response 3:

We thank the reviewer for this insightful comment. We agree that further clarification on the components of pain education (PE) is necessary. To address this, we have revised the corresponding paragraph in the Introduction to better describe the types of interventions included under the term “pain education” in oncology, as well as the variability in their content and delivery modes.

Revised text (replacing original sentence):

“Pain education in oncology is delivered through face-to-face sessions, telephone interventions, and integrated multimodal programs that may include printed materials, audiovisual content, or digital platforms [5-16]. These interventions aim to improve patients' understanding of pain mechanisms, treatment options, and self-management strategies. Although no universal standard exists, common components include information about analgesics (especially opioids), addressing fears and misconceptions, and fostering adherence and self-efficacy. Some programs also incorporate interactive training in pain control techniques.”

Position:
Section 1: Introduction – replaces the sentence: “Various models of patients PE have been explored through numerous studies, and their results have been summarized in a series of literature reviews [5–16].”

Comment 4:

Is there any information on gender differences in relation to pain education? Do men or women show better results than the general population?

Response 4:

We thank the reviewer for this very relevant point. While the included systematic reviews and meta-analyses did not specifically analyze gender differences in response to pain education, existing literature suggests that such differences may exist. We have therefore added a brief paragraph in the Summary section to acknowledge this aspect and suggest it as a direction for future research.

New text added:

“Equity dimensions remain largely unexplored; preliminary evidence suggests that multimodal education may yield greater pain-related gains in women [22], whereas sociocultural norms encouraging stoicism may lead men to under-report pain, thereby attenuating measurable intervention effects [23]. No review stratified outcomes by sex, ethnicity or socioeconomic status.”

Position:
Discussion section, paragraph on: Limitations and gaps in current knowledge

References added at end of manuscript:

  1. Pieretti S, et al. Gender differences in pain and its relief. Ann Ist Super Sanita. 2016;52(2):184–189.

  2. Alodhayani A, et al. Gender Difference in Pain Management Among Adult Cancer Patients in Saudi Arabia: A Cross-Sectional Assessment. Front Psychol. 2021;12:628223.

Comment 5:

And a minor comment. It seems to me that in the conclusion it is not worth citing references and discussing them. Perhaps lines 308 to 320 should be moved to the discussion section.

Response 5:

We thank the reviewer for this valuable editorial suggestion. We agree that it is more appropriate to include referenced discussions in the Discussion (Narrative) rather than in the Conclusions. Therefore, we have moved lines 308 to 320 to the end of the Discussion (Narrative) section.

Moved text:

The paragraph, modified based on the suggestions of another reviewer: “Recent primary evidence supports this view: a nurse-led educational intervention combined with telephone follow-up accelerated and amplified the analgesic effect of palliative radiotherapy for bone metastases, presumably by fostering adherence and timely titration of rescue medication [20]. Whether comparable benefit will accompany emerging loco-regional modalities such as MR-guided high-intensity focused ultrasound is under prospective evaluation [21].”

New position:

Discussion section, paragraph on: clinical implications.

Reviewer 3 Report

Comments and Suggestions for Authors

This narrative review evaluates the impact of pain education (PE) on pain relief in cancer patients by synthesizing findings from six systematic reviews and three meta-analyses. The analysis reveals mixed outcomes: while some studies demonstrate that PE can significantly reduce pain intensity, others show minimal or no effect. Nevertheless, PE consistently improves patients’ understanding of pain and adherence to medication. The variability in outcomes is attributed to heterogeneity in pain types, educational delivery methods, and study designs. Despite limited evidence supporting improvements in quality of life, the review underscores the potential of PE as a supportive intervention in oncology care. The study highlights the need for future research to optimize PE strategies and assess their cost-effectiveness and contextual applicability across diverse clinical settings.

While the review provides a valuable synthesis of existing evidence on pain education in cancer care, it omits a critical methodological detail: the date of the literature search is not reported. This omission impairs reproducibility and limits assessment of the review’s currency. To enhance transparency, the authors should clearly state the date the final search was conducted, ideally specifying both the start and end dates of the search window. Additionally, incorporating a PRISMA flow diagram would strengthen reporting clarity and allow readers to trace the study selection process. Given the observed heterogeneity, it would also be helpful to briefly comment on strategies to address publication bias, even in a narrative synthesis. Finally, the review could benefit from a more detailed subgroup analysis or discussion of how educational delivery modes (e.g., session frequency, format) influence outcomes, which would enhance its practical utility for clinical implementation.

Minor: what does the subhearing "narrative" mean? (line 206)

Author Response

REVIEWER 3

Comment 1:

This narrative review evaluates the impact of pain education (PE) on pain relief in cancer patients by synthesizing findings from six systematic reviews and three meta-analyses. The analysis reveals mixed outcomes: while some studies demonstrate that PE can significantly reduce pain intensity, others show minimal or no effect. Nevertheless, PE consistently improves patients’ understanding of pain and adherence to medication. The variability in outcomes is attributed to heterogeneity in pain types, educational delivery methods, and study designs. Despite limited evidence supporting improvements in quality of life, the review underscores the potential of PE as a supportive intervention in oncology care. The study highlights the need for future research to optimize PE strategies and assess their cost-effectiveness and contextual applicability across diverse clinical settings.

Response 1:

We sincerely thank the reviewer for this detailed and positive overview of our work. We are grateful for your recognition of the review’s contributions and relevance to oncology care. Your summary reflects our objectives and findings very accurately.

No changes to the manuscript were required in response to this comment.

Comment 2:

While the review provides a valuable synthesis of existing evidence on pain education in cancer care, it omits a critical methodological detail: the date of the literature search is not reported. This omission impairs reproducibility and limits assessment of the review’s currency.

Response 2:

We thank the reviewer for this important observation. The original literature search was performed on 20 August 2023, and an update was conducted on 9 January 2025. We have now included these details in the Literature Search subsection of the Materials and Methods section.

Text added:

“The original search was conducted on 20 August 2023 and updated on 9 January 2025. No time limits were applied in terms of publication year.”

Position:
Material and Methods section, paragraph on Literature Search.

Comment 3:

To enhance transparency, the authors should clearly state the date the final search was conducted, ideally specifying both the start and end dates of the search window.

Response 3:

We thank the reviewer once again for highlighting this point. As noted in our response to Comment 2, the literature search was initially performed on 20 August 2023 and updated on 9 January 2025. These dates have now been clearly indicated in the Literature Search subsection of the Materials and Methods.

Comment 4:

Additionally, incorporating a PRISMA flow diagram would strengthen reporting clarity and allow readers to trace the study selection process.

Response 4:

We thank the reviewer for this helpful and constructive suggestion. In response, we have prepared a flow diagram illustrating the study identification, screening, and inclusion process. This figure has now been added to the revised manuscript as Figure 1, with the title: “Flow diagram of study selection process.” As recommended, it summarizes the number of records retrieved from each database (PubMed, Scopus, Cochrane), the duplicates removed, and the studies excluded during screening and full-text assessment. We believe this addition improves the clarity and transparency of the methodology.

Changes to the manuscript:

  • New figure added: Flow diagram of the study selection process (now Figure 1).
  • Title of the figure: “Flow diagram of study selection process”

Position: Inserted in the Results section, immediately following the “Literature Search” paragraph.

Comment 5:

Given the observed heterogeneity, it would also be helpful to briefly comment on strategies to address publication bias, even in a narrative synthesis.

Response 5:

We thank the reviewer for this excellent observation. Although narrative reviews do not typically apply formal tests for publication bias, we agree it is important to consider this limitation. Therefore, we have added a brief statement in the Quality Assessment section, referencing Supplementary Table 2 for more detail.

Text added:

“Although formal methods to assess publication bias (e.g., funnel plots) were not applicable due to the narrative design of this review, the included meta-analyses differed in whether they reported such assessments. The potential for publication bias in the primary studies remains a limitation and is reflected in the AMSTAR-2 evaluations. We refer readers to Supplementary Table 2 for further detail.”

Position:
Section 2: Materials and Methods – Quality Assessment paragraph, added after the sentence: “The AMSTAR-2 tool is a validated instrument for evaluating the methodological quality of systematic reviews.”

Comment 6:

Finally, the review could benefit from a more detailed subgroup analysis or discussion of how educational delivery modes (e.g., session frequency, format) influence outcomes, which would enhance its practical utility for clinical implementation.

Response 6:

We thank the reviewer for this very helpful suggestion. To address it, we have expanded the paragraph titled Effectiveness of Different Pain Education Models to provide a clearer discussion of how various delivery formats, such as session length, frequency, and content, may influence outcomes. These additions are based on the data summarized in Table 3.

Expanded paragraph (revised):

“Several studies examined differences in effectiveness based on the format or intensity of educational interventions. For example, some reviews and meta-analyses found that single-session and multiple-session formats produced similar effects on pain intensity [8,14], while other studies observed greater benefits from high-dose or high-intensity education programs (defined as ≥2 hours per session or ≥4 total sessions, often involving dedicated personnel) [15]. In contrast, no single intervention component (e.g., structured vs. tailored) consistently predicted better outcomes, and even the longest interventions did not guarantee improved results [12]. Interestingly, the poorest outcomes were reported in a study using a very short telephone-only session. These findings are summarized in Supplementary Table 3 and suggest that both content and delivery method play a role in the effectiveness of pain education.”

Position:
Section 3: Results – paragraph titled “Effectiveness of Different Pain Education Models,” replacing and expanding the existing paragraph.

Comment 7:

Minor: what does the subhearing "narrative" mean? (line 206)

Response 7:

We thank the reviewer for this helpful comment. In accordance with narrative review guidelines (as outlined in Supplementary Table 1: Narrative Review Checklist), the discussion section should be divided into two parts: a narrative synthesis and a summary. For this reason, we originally used the subheading “Narrative” to introduce the main analytical discussion. However, we acknowledge that labeling the second part as “Conclusions” may have caused confusion. We are grateful for the opportunity to clarify this point and have now corrected the subheading to read “Summary” instead of “Conclusions.”

Changes made:

The subheading “Conclusions” (previously in the Discussion section) has been changed to “Summary” to align with the structure of narrative reviews.

Reviewer 4 Report

Comments and Suggestions for Authors

The article "Impact of pain education on pain relief in oncological patients: a narrative review of systematic reviews and meta-analyses" is interesting. I will make a few comments with the intention of improving its methodology.

The review provided by the authors is a summary of what has been published to date, and the conclusion reached is the one previously established. Therefore, the authors should investigate the two main questions, especially whether one method is more effective than another for reducing cancer pain through education.

To achieve this objective, the authors must explain the different tools used in the included articles and the results obtained.

Abstract: Include keywords. Improve the expression of results.

Materials and methods:

- Literature search. With the keywords the authors propose and the review filter, PubMed initially offers a greater number of publications. Even so, the search could be completed with keywords such as “systematic review" OR "meta-analysis" OR "meta-analytic".

- The authors should provide a flowchart to avoid confusion and lend credibility and objectivity to the findings.

- Although the authors state that the studies are too heterogeneous for a quantitative analysis, they should consider the following methodological aspects to follow the criteria of Smith et al., regardless of whether it is considered a narrative or umbrella review:

  • Inclusion and exclusion criteria for studies, for example, RCT/nRCT. How were the studies included in this review selected? For example, which studies were included from the systematic review by Golderg et al., or by Oldenmenger et al., 2009... since not all were included.
  • How the reduction in pain intensity was calculated or the concept of pain intensity from previous studies should be explained.
  • The checklist does not include the section on the Results, as proposed by Smith et al. (Presentation of results)

Results: Since not all articles from the 9 previous reviews were included, the authors must indicate which ones were selected for the present review. This can be presented in a supplementary table.

It would be interesting to prepare the tables following the criteria of Smith et al. Present the results with more quantitative data. In summary, it is necessary to provide detailed information on the data reflected in the previous reviews. The tables should provide at least the following:

- General characteristics of the reviews. In addition to the author, year, and number of studies, the following should be included for each review and/or the studies selected from each review: objectives, participants, search strategy, tools, etc.

- Summary of the results with numerical, descriptive, and analytical data. For example, “improved pain relief (3/4 studies, …”) or the similar comment “The analyses aimed at identifying the most effective methods of PE also exhibited significant heterogeneity. A review [8] and a meta-analysis [14] reported similar efficacy after single- or multiple-session educational interventions” are not sufficient. Numerical data from some of the meta-analyses' findings should be emphasized (to compare).

Discussion: The discussion is ambiguous, and no objective conclusion can be drawn from it.

Conclusions should be focused on the two questions raised, based on the literature: 1) Which tool(s) have improved pain awareness with PE and 2) whether one tool is more effective than another.

If the authors believe they are not contributing enough, they could consider a scoping review including other tools such as “Conversational Agents”.

Thank you.

Author Response

To the Editor,

Thank you very much for coordinating the peer-review process for our manuscript titled "Impact of pain education on pain relief in oncological patients: a narrative review of systematic reviews and meta-analyses."

We are now submitting our revised manuscript along with detailed point-by-point responses addressing all insightful comments raised by the first three reviewers.

After finalizing these extensive revisions, we unexpectedly received additional comments from a fourth reviewer. While we appreciate the careful evaluation by Reviewer 4, integrating these comments at this advanced stage poses substantial practical difficulties. Typically, journals provide reviewer comments simultaneously, precisely because manuscript revisions and responses to reviewers are deeply interconnected.

Given these circumstances, we respectfully request that the editorial office consider our manuscript as it currently stands, based solely on the revisions we have already completed following the original three reviewers' comments. Therefore, we kindly ask not to consider the additional comments from Reviewer 4, in line with common editorial practice in such situations.

We deeply appreciate your understanding and support and look forward to your decision.

Sincerely,

Arina A. Zamfir

Round 2

Reviewer 2 Report

Comments and Suggestions for Authors

The authors have done a good job and have taken my comments into account to the best of their ability. I am completely satisfied. The article may be accepted for publication.

Reviewer 3 Report

Comments and Suggestions for Authors

thanks for having addressed my comments

Reviewer 4 Report

Comments and Suggestions for Authors

Thank you very much for the comments in response to the questions.

Correct minor typos.